# Research of Online Hand–Eye Calibration Method Based on ChArUco Board

**DOI:** 10.3390/s22103805

**Published:** 2022-05-17

**Authors:** Wenwei Lin, Peidong Liang, Guantai Luo, Ziyang Zhao, Chentao Zhang

**Affiliations:** Fujian (Quanzhou)-HIT Research Institute of Engineering and Technology, Quanzhou 362000, China; lww2127@hitqz.com (W.L.); lpd0004@hitqz.com (P.L.); lgt2091@hitqz.com (G.L.); zzy2103@hitqz.com (Z.Z.)

**Keywords:** hand–eye calibration, ChArUco board, robot

## Abstract

To solve the problem of inflexibility of offline hand–eye calibration in “eye-in-hand” modes, an online hand–eye calibration method based on the ChArUco board is proposed in this paper. Firstly, a hand–eye calibration model based on the ChArUco board is established, by analyzing the mathematical model of hand–eye calibration, and the image features of the ChArUco board. According to the advantages of the ChArUco board, with both the checkerboard and the ArUco marker, an online hand–eye calibration algorithm based on the ChArUco board is designed. Then, the online hand–eye calibration algorithm, based on the ChArUco board, is used to realize the dynamic adjustment of the hand–eye position relationship. Finally, the hand–eye calibration experiment is carried out to verify the accuracy of the hand–eye calibration based on the ChArUco board. The robustness and accuracy of the proposed method are verified by online hand–eye calibration experiments. The experimental results show that the accuracy of the online hand–eye calibration method proposed in this paper is between 0.4 mm and 0.6 mm, which is almost the same as the offline hand–eye calibration accuracy. The method in this paper utilizes the advantages of the ChArUco board to realize online hand–eye calibration, which improves the flexibility and robustness of hand–eye calibration.

## 1. Introduction

The application of industrial robots has greatly improved the production efficiency and product quality of enterprises [1,2]. However, the traditional working methods of industrial robots have been unable to meet the increasing production demand. Industrial robots equipped with vision sensors perceive the surrounding environment through image processing technology, which improves the “flexibility” of robot operations, and enables robots to complete more complex production tasks [3,4,5]. Hand–eye calibration is an indispensable part of the visual servo operation of the robot, which is an important bridge between the robot and the vision sensor. Hand–eye calibration is the process of solving the coordinate transformation relationship between the camera coordinate system and the robot coordinate system. The accuracy of hand–eye calibration directly affects the accuracy of the robot operation. For the research of hand–eye calibration, scholars have proposed many theories and methods. Wu et al. [6] and Yang et al. [7] solved the hand–eye problem by tracking the position of the calibration ball in 3D camera and the movement trajectory of the robot. Chen et al. [8] and Wang et al. [9] calibrated the relationship of the position between the camera and the calibrator by identifying the feature points of the 3D calibration object, and solved the hand–eye matrix through the corresponding posture of the robot. Although the hand–eye calibration method based on 3D calibrators is one of the most effective ways to solve the hand–eye matrix, the processing of 3D data consumes a lot of time, and the quality of the 3D point cloud has absolute influence on the calculation results.

Compared with the hand–eye calibration method based on the 3D calibrators, the hand–eye calibration method based on 2D images is more widely used. Since the feature points of the checkerboard pattern are easy to detect, most of the classical hand–eye calibration theories use it as the calibrator [10,11,12]. In recent years, other scholars have carried out research on hand–eye calibration methods based on the checkerboard. Deng et al. [13] proposed a hand–eye calibration method based on a unit octonion, and verified the effectiveness of the method using a camera and a checkerboard. Lee et al. [14] designed an automatic hand–eye calibration system based on a checkerboard calibration. In addition, as an image coding technique, an ArUco marker is often used for hand–eye calibration. Huang et al. [15] adopted the hand–eye calibration method, based on using ArUco markers and the support vector machine (SVM) detection method, to identify specific objects to realize flexible grasping of the robot. Feng et al. [16] realized the robot’s autonomous assembly and scanning was based on vision guidance, by using the hand–eye calibration method based on the ArUco marker. The black and white interlaced pattern features of the checkerboard make the feature points easy to identify. However, when the checkerboard target is occluded, or the lighting conditions are poor, the pose of the calibration board is difficult to recognize. ArUco markers have the advantages of rapid detection and flexibility. However, the detection accuracy of the ArUco marker in corners is not very high. Furthermore, most hand–eye calibration algorithms are performed offline. When the relative position of the camera and the robot changes, the hand–eye calibration often needs to be reperformed.

To solve these problems, an online hand–eye calibration method based on the ChArUco board is proposed in this paper. The ChArUco board is a combination of an ArUco marker and a checkerboard, and possesses the advantages of both. The ChArUco board rectifies the shortcomings of the ArUco marker’s poor positioning accuracy by using the high sub-pixel detection accuracy of the checkerboard corners of the calibration board, and solves the problem of the poor anti-interference of the checkerboard through the unique encoding of the ArUco marker. Based on the detection advantages of the ChArUco board, this paper designs a closed-loop feedback adjustment system for the robot to realize online hand–eye calibration of eye-in-hand mode.

The other parts of the paper are arranged as follows: Section 2 introduces the principle of hand–eye calibration and the ChArUco board; Section 3 describes the online hand–eye calibration method, based on the ChArUco board; Section 4 verifies the accuracy and feasibility of the method proposed in this paper; finally, Section 5 summarizes the work of this paper, and looks into future research issues.

## 2. Hand–Eye Calibration and ChArUco Board

The purpose of hand–eye calibration was to calculate the coordinate transformation relationship between the camera coordinate system, and the robot coordinate system, that is, to solve the rotation matrix ***R*** and the translation vector ***t***. The essence of hand–eye calibration was to solve the problem of AX = XB [17,18,19]. Hand–eye calibration was divided into eye-in-hand and eye-to-hand modes, as shown in Figure 1. As shown in Figure 1b, the relative position of the robot base and the calibration board was unchanged, as was the relative position of the camera and the robot end. According to multiple sets of known invariants, the hand–eye calibration matrix could be solved.

As shown in Formula (2), the hand–eye calibration matrix could be solved through the calculation of the robot pose transformation, and the camera extrinsic parameters.
(1)Tg(1)b⋅Tcg⋅Tt(1)c=Tg(2)b⋅Tcg⋅Tt(2)c
(2)(Tg(2)b)−1⋅Tg(1)b⋅Tcg=Tcg⋅Tt(2)c⋅(Tt(1)c)−1

Let (Tg(2)b)−1⋅Tg(1)b=A, Tt(2)c⋅(Tt(1)c)−1=B, Tcg=X, then:(3)AX=XB
where Tgb represents the homogeneous matrix of the robot end coordinate system, relative to the robot base coordinate system; Ttc represents the homogeneous matrix of the calibration board coordinate system, relative to the camera coordinate system; Tcg represents the homogeneous matrix of the camera coordinate system, relative to the robot end coordinate system.

According to the described mathematical model of hand–eye calibration, the solution process of hand–eye calibration needed to obtain the homogeneous matrix Ttc of the calibration board coordinate system, relative to the camera coordinate system, that is, it needed to calculate the external parameters of the camera. In this paper, the ChArUco board was selected as the calibration object to calculate Ttc.

ArUco markers have the advantages of flexibility and easy detection. However, the ArUco marker has the problem that the detection accuracy of the edge corners is not high. Even if sub-pixel processing is performed on the corner, the expected accuracy is still not achieved. The black and white interlaced pattern of the checkerboard makes the corners easy to detect. Unfortunately, the flexibility of the checkerboard is not as extensive as the ArUco marker. When the checkerboard was used as a calibration object, the checkerboard needed to be completely visible, and could be blocked. The ChArUco board possessed the advantages of both the checkerboard and the ArUco marker. In addition, the ChArUco board ameliorated the deficiencies of both. Figure 2 shows a schematic diagram of three calibration objects.

## 3. Online Hand–Eye Calibration Based on ChArUco Board

As shown in Figure 3, when corner detection was performed on the ChArUco board, the corners of the checkerboard and the coding pattern were identified. According to the decoding information of the coding pattern, the positions of each corner of the calibration board could be sorted in an orderly manner. Even if the ChArUco board was partially occluded, it did not affect the order of the corners.

As shown in Figure 4, it is the calculation result of camera extrinsic parameters under different occlusion conditions of the ChArUco board. When the occluded area of the ChArUco board was small, it did not affect the calculation results of the camera external parameters. When the occluded area of the ChArUco board was too large, the external parameters of the camera could not be calculated. However, some coding patterns and checkerboard positions could still be recognized.

In addition, the pose of each recognizable encoding pattern on the ChArUco board could be computed as if the pose of a single ArUco marker were recognized. Figure 5a shows the pose of the ArUco marker in the camera coordinate system. Figure 5b shows the pose of each recognizable coding pattern of ChArUco in the camera coordinate system.

Based on the detection advantages of the ChArUco board, this paper designed a closed-loop feedback adjustment system for the robot to realize online hand–eye calibration. As shown in Figure 6, in the case of the eye-in-hand, the camera could be seen as a “tool” attached to the end of the robot. The “inaccurate” Tcg was calculated by the robot-based tool calibration method [20]. In addition, Ttc could be calculated by the detection of the ChArUco board. Tgb could be obtained directly through the robot teach pendant. With the above information, the approximate positional relationship between the ChArUco board and the robot could be calculated.

Figure 7 shows the online hand–eye calibration process, based on the ChArUco board. Firstly, after the robot moved to the teaching posture, the camera captured the ChArUco board calibration image. Then, it was judged whether the position of the calibration board satisfied the condition by detecting the ChArUco board. If it was not satisfied, the robot posture was automatically adjusted, according to the feedback of the ChArUco board position information, and the above steps were repeated. Otherwise, the data were saved, the posture of the robot adjusted, and the next teaching action was reached. When the collected hand–eye calibration data met the requirements, the hand–eye calibration matrix was solved.

Figure 8, shows a schematic diagram of the image changes before and after the robot feedback adjustment.

## 4. Experiments and Analysis

In this paper, Hikvision’s MV-CE050-30GM camera, FUJINON’s CF8ZA-1S lens and ABB’s IRB2600-20 robot were used to build the experimental platform. The experimental platform is shown in Figure 9. The camera was fixed on the end of the robot, and the calibration board was fixed on the experimental table. Based on this experimental platform, this paper tested the accuracy of the three calibration methods of the ArUco marker, the checkerboard and the ChArUco board, and carried out an online hand–eye calibration experiment based on the ChArUco board.

Before the experiment, the internal parameters of the camera used in the experiment were calibrated. The calculation results of camera internal parameters were as follows:$$\mathrm{CameraMatrix}=\left[\begin{array}{ccc} 3797.467 & 0 & 1310.649\\ 0 & 3801.126 & 923.535\\ 0 & 0 & 1 \end{array}\right]$$
$$\mathrm{DistCoeffs}=\left[\begin{array}{ccccc} -0.1793 & 1.2373 & -0.0014 & 0.0008 & -7.1253 \end{array}\right]$$

In this paper, 10 sets of hand–eye calibration independent experiments were carried out, and 25 sets of hand–eye calibration data, based on the ArUco marker, the checkerboard and the ChArUco board, were collected in each experiment. The calculation results of camera extrinsic parameters for the three calibration methods are shown in Figure 10.

Figure 11 shows the hand–eye calibration uncertainty, based on the ArUco marker, the checkerboard and the ChArUco board. From the analysis of the experimental data, it could be seen that the hand–eye calibration accuracy of the ArUco marker was the worst, and compared with the checkerboard and the ChArUco board, the maximum deviation was 1.5 mm. Compared with the checkerboard and the ChArUco board, the calibration accuracy of the two was almost the same.

In online hand–eye calibration experiments based on the ChArUco board, 10 sets of online hand–eye calibration independent experiments were carried out. In a single set of independent experiments, 25 sets of robot teaching actions were set, and the ChArUco board was placed away from the center of the camera’s field of view. Figure 12 shows the change of the camera angle of view, before and after the online hand–eye calibration experiment based on the ChArUco board. Through the code identification of the ArUco marker on the ChArUco board, and the statistics of the identified code number, the position of the ChArUco board and its proportion in the image could be determined. It can be seen from the figure, that the posture recognition of the ChArUco board could be realized after the robot adjustment, through closed-loop feedback.

In addition, as shown in Figure 13, this paper conducted uncertainty analysis on 10 sets of online hand–eye calibration experiments. It can be seen from the figure, that the uncertainty of the online hand–eye calibration, based on the ChArUco board, is between 0.4 mm and 0.6 mm, which is almost the same as the accuracy of the offline hand–eye calibration. The experimental results showed that the method in this paper effectively solved the problem of online hand–eye calibration, and also ensured the stability of the calibration accuracy.

Based on the above experimental results, it can be seen that the online hand–eye calibration method proposed in this paper could effectively solve the problem of online hand–eye calibration, and had good performance in robustness and accuracy.

## 5. Conclusions

In order to reflect the practicability of this method, the performance of this method was compared with the existing methods, and the results are shown in Table 1. The methods of [21,22] could only obtain the position of the calibration plate in the image. In the process of hand–eye calibration, the methods of [21,22] needed to ensure that the calibration plate was visible as a whole, and the checkerboard feature points and circular patterns could not be blocked. The ChArUco board used in this method could obtain the position and posture of the calibration board, and had certain interference to the occlusion situation. Therefore, the method in this paper had better flexibility in robot adjustment. In the process of hand–eye calibration, the method in this paper had better robustness. The feature point calculation process of the ChArUco board used in this paper was more time-consuming, but the method in this paper was able to obtain the position and attitude of the calibration board, which reduced the number of adjustments of the robot. The overall time consumption was better than the methods of [21,22]. The method in this paper only needed 0.5–2 s to complete a single valid hand–eye calibration datum. In addition, the method in this paper had better flexibility and robustness, so that more effective hand–eye calibration data could be collected under the same disturbance. Therefore, the accuracy of hand–eye calibration would also be better.

Aiming at the problem of inflexible offline hand–eye calibration in eye-in-hand mode, an online hand–eye calibration method based on the ChArUco board was proposed in this paper. The method in this paper utilized the advantages of the ChArUco board, which has the advantages of high sub-pixel recognition accuracy of the checkerboard corner, and the strong flexibility of the ArUco marker, to realize the positioning of the calibration board. The position relationship between the calibration board and the robot was established. Then, the closed-loop feedback automatically adjusted the robot by detecting the position of the ChArUco board in the image. Enough hand–eye calibration data were collected by robot automatic control to complete online hand–eye calibration. In this paper, the accuracy of hand–eye calibration based on the ChArUco board was verified by comparative experiments. The robustness and accuracy of the method were verified by online hand–eye calibration experiments.

In the current research of this paper, the influence of the hand–eye calibration caused by the robot motion error was not considered. In future work, we will study the influence of robot motion error on hand–eye calibration accuracy, and consider how to eliminate the influence of robot motion error. This will be an interesting and meaningful research direction.

## Figures and Tables

**Figure 1 sensors-22-03805-f001:**
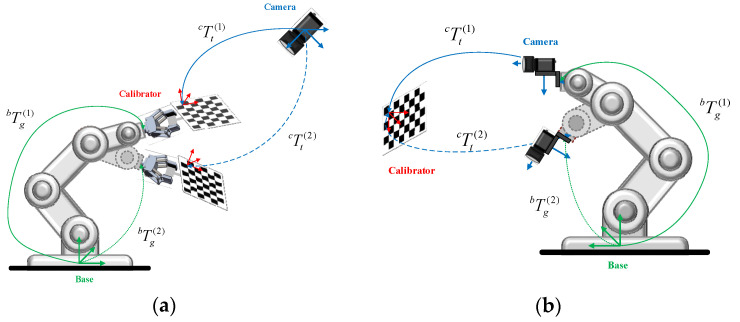
Two hand–eye calibration modes: (**a**) Eye-to-hand; (**b**) Eye-in-hand. Where Base represents the base coordinate system of the robot; Camera represents the camera coordinate system; and Calibrator represents the calibration board.

**Figure 2 sensors-22-03805-f002:**
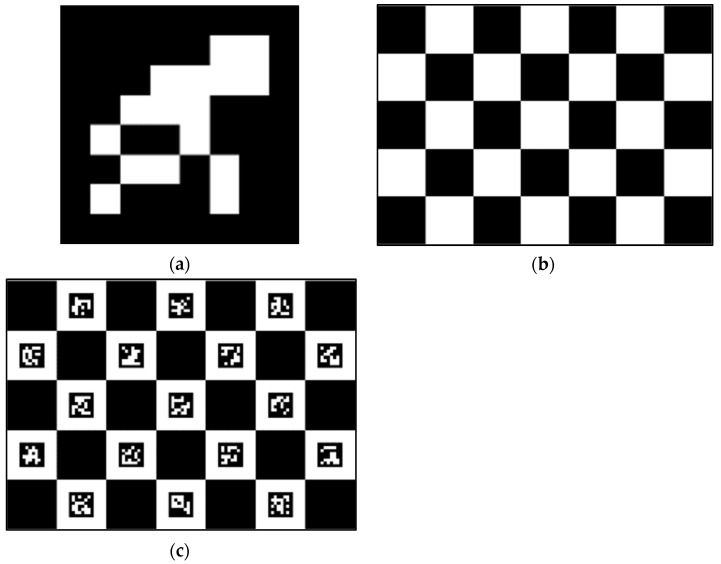
(**a**) ArUco marker; (**b**) Checkerboard; and (**c**) ChArUco board.

**Figure 3 sensors-22-03805-f003:**
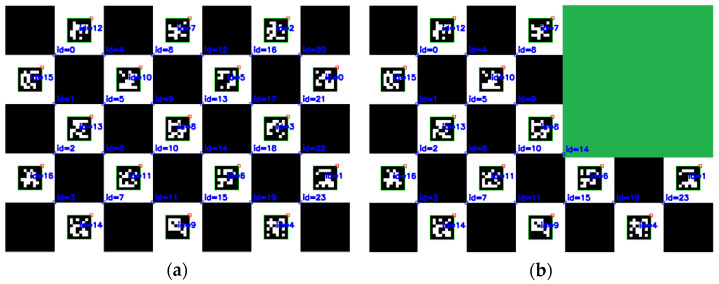
ChArUco board corner detection under different circumstances: (**a**) Corner detection under normal circumstances; (**b**) Corner detection when corners are occluded; and (**c**) Corner detection with intermediate occlusion.

**Figure 4 sensors-22-03805-f004:**
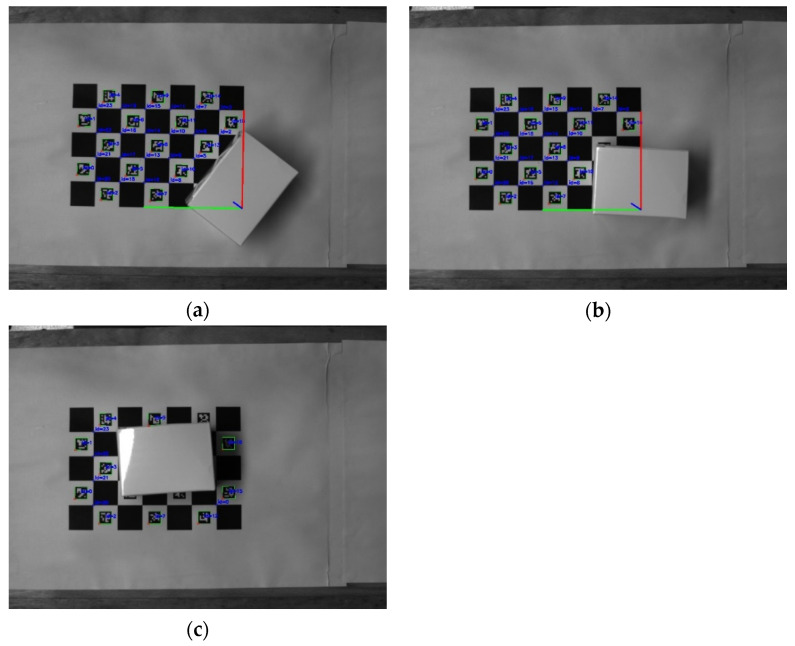
Calculation results of camera extrinsic parameters based on ChArUco board, under different occlusion conditions: (**a**,**b**) Calculation results of camera extrinsic parameters for small area occlusion; (**c**) Calculation results of camera extrinsic parameters for large area occlusion.

**Figure 5 sensors-22-03805-f005:**
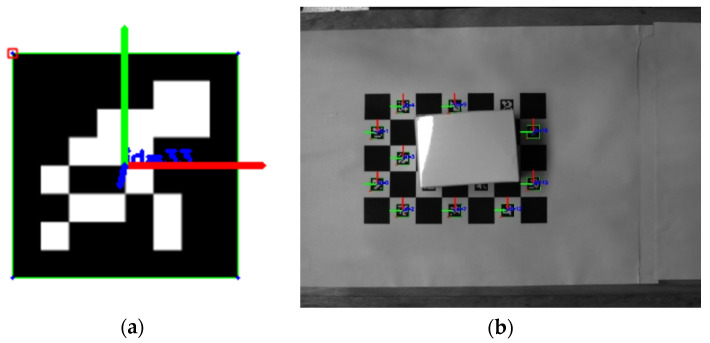
(**a**) The pose of the ArUco marker in the camera coordinate system; (**b**) The pose of each recognizable coding pattern of the ChArUco board in the camera coordinate system.

**Figure 6 sensors-22-03805-f006:**
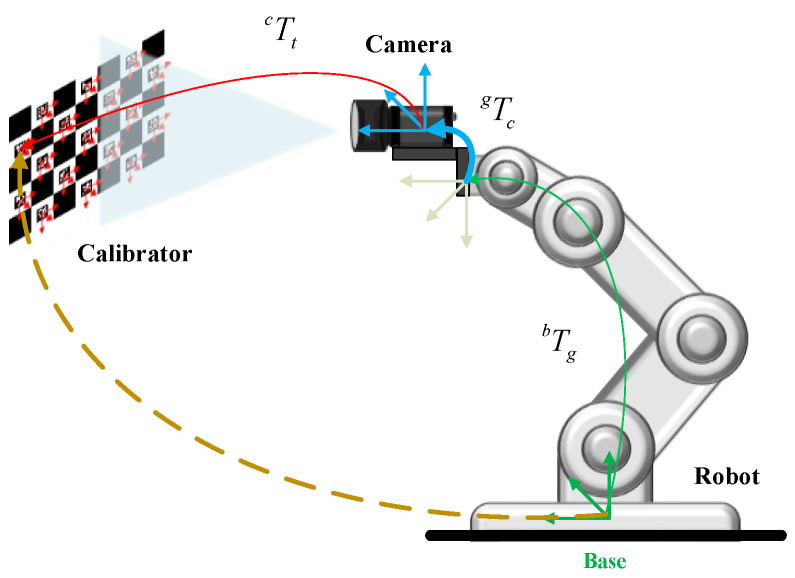
Schematic diagram of the positional relationship between the ChArUco board and the robot.

**Figure 7 sensors-22-03805-f007:**
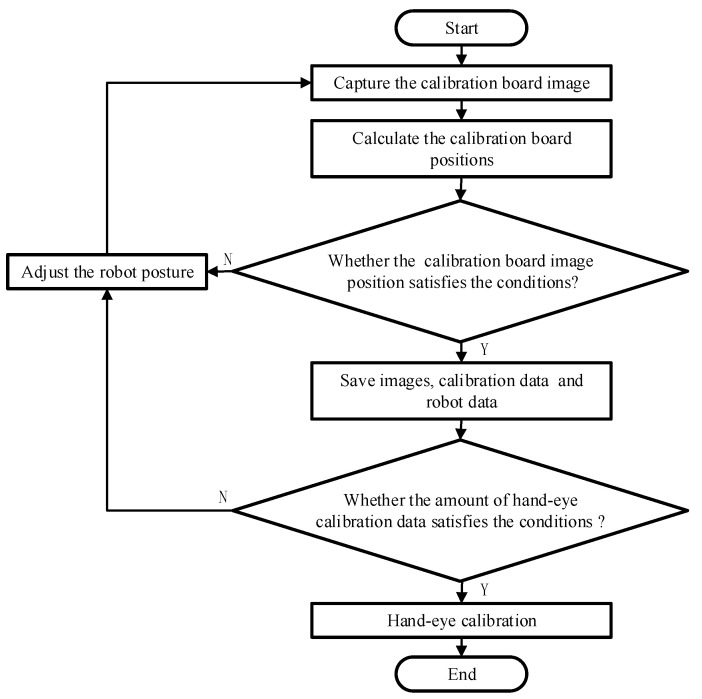
Process of online hand–eye calibration based on ChArUco board.

**Figure 8 sensors-22-03805-f008:**
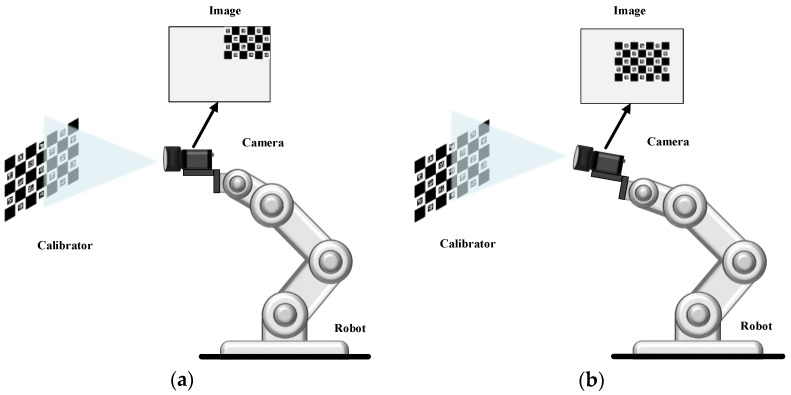
(**a**) Image captured by camera before robot adjustment; (**b**) Image captured by camera after robot adjustment.

**Figure 9 sensors-22-03805-f009:**
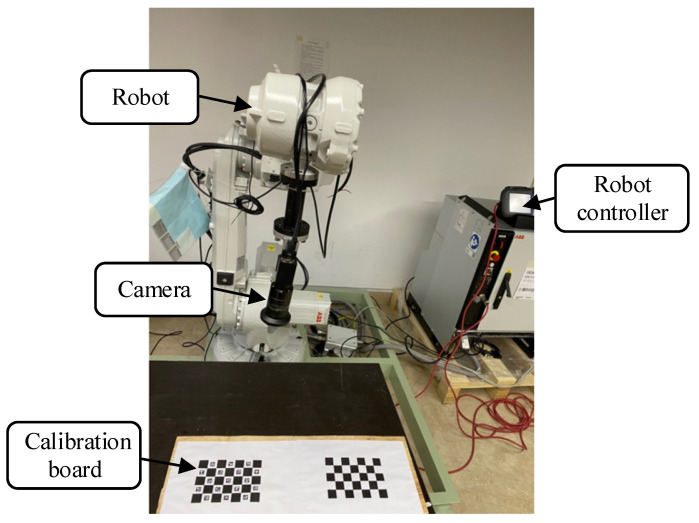
Hand–eye calibration experimental platform.

**Figure 10 sensors-22-03805-f010:**
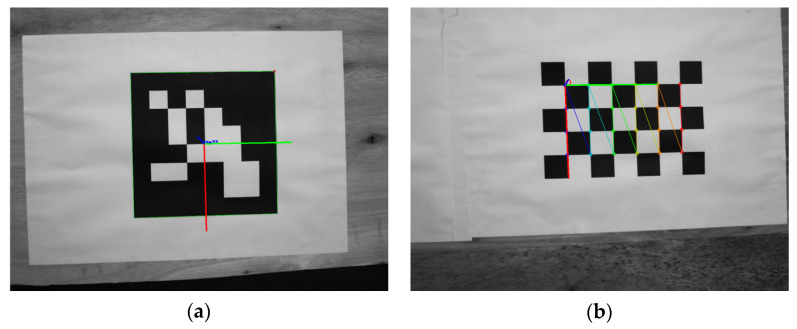
(**a**) Calculation of camera external parameters based on ArUco Marker; (**b**) Calculation of camera external parameters based on checkerboard; and (**c**) Calculation of camera external parameters based on ChArUco board.

**Figure 11 sensors-22-03805-f011:**
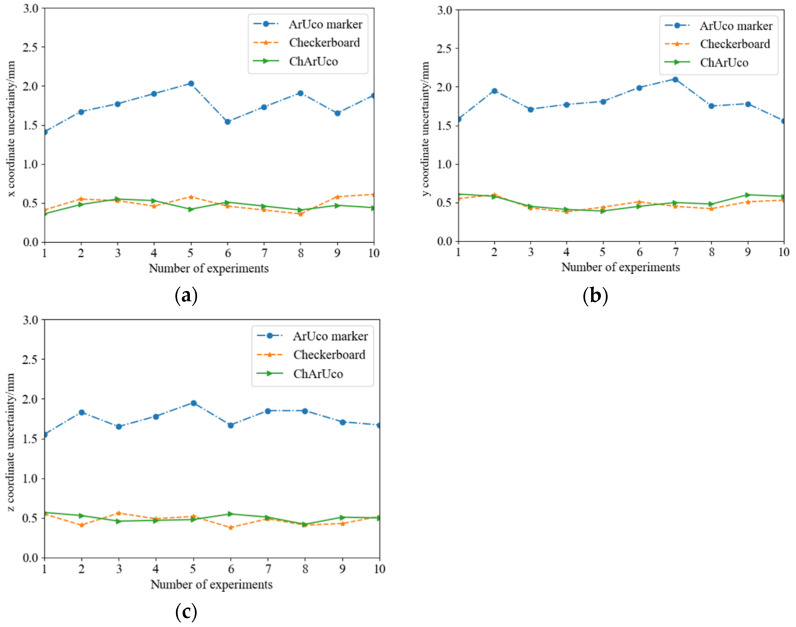
Hand–eye calibration uncertainty: (**a**) x coordinate reprojection uncertainty; (**b**) y coordinate reprojection uncertainty; and (**c**) z coordinate reprojection uncertainty.

**Figure 12 sensors-22-03805-f012:**
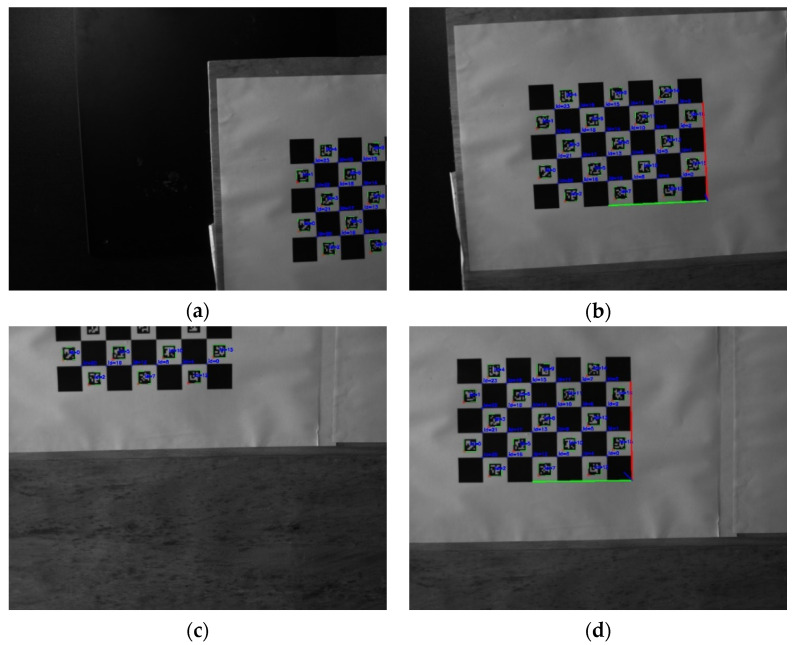
Online hand–eye calibration experiment, based on ChArUco board: (**a**,**c**) Image captured by camera before robot adjustment; (**b**,**d**) Image captured by camera after robot adjustment.

**Figure 13 sensors-22-03805-f013:**
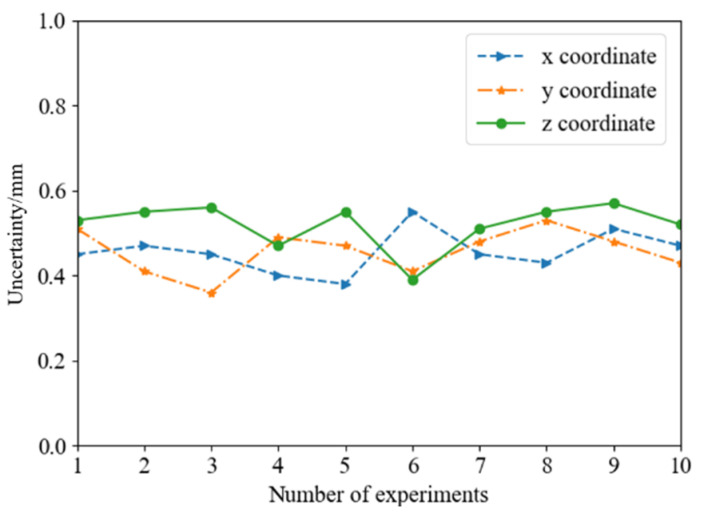
Analysis of online hand–eye calibration uncertainty.

**Table 1 sensors-22-03805-t001:** Performance comparison of online hand–eye calibration algorithms.

Methods	Flexibility	Robustness	Time Consumption	Accuracy
Reference [21]	Low	Bad	High	Low
Reference [22]	Low	Bad	High	Low
Our Method	High	Good	Low	High

## Data Availability

The data presented in this study are available on request from the corresponding author. The data are not publicly available due to privacy.

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
