# Peer review of "Research of Online Hand–Eye Calibration Method Based on ChArUco Board"

_sensors, 2022, doi:10.3390/s22103805_

Round 1

Reviewer 1 Report

The research issue is very relevant. The idea of using ArUco markers and checkerboard for hand-eye calibration is quite interesting.

There are some comments:

1) The authors should add information about the inertia (time delay) of the online hand-eye calibration of the robot.

2) The authors should make a table with the technical parameters of the proposed system. This table will be useful for comparison with existing calibration systems.

3) In the Conclusions, the authors should explain why the proposed system is better than the known ones using the existing systems as examples.

4) "Measurement error" is an obsolete term. The error evaluation methodology should follow the uncertainty approach as introduced in "GUM: Evaluation of measurement data - Guide to the expression of uncertainty in measurement). I suggest using "uncertainty" instead of "error" in Figures 11 and 13 on the labels of the vertical axis and curves. The same changes should be made in the captions of Figures 11 and 13 and in the text on pages 9-12.

5) Please, place a caption under Figure 13 at the bottom of page 10.

Author Response

Thanks to the reviewers for their valuable comments on the manuscript, we have revised the manuscript based on your comments. For details, please see the attachment and the revised manuscript.

Reviewer 2 Report

1. What is the robot based tool calibration method mentioned for gTc matrix calculation?

2. It should be more explicit and point out in the document from the abstract that the design used is eye in hand.

3. You could place a Black and white pixel ratio data table to know in what ratio the robot's camera detects.

4. Specify in the introduction that the configuration of the camera and the robot is eye in hand.

Author Response

(The authors gave the same response as above.)

Reviewer 3 Report

Very good and clear language. Extremely easy to read, follow and understand. Very interesting in terms of applicability. Can be of great interest to engineers working with robots - considerable/moderate citability expected. (Although it doesn't feel like a "scientific" paper.)

Good work, however I would love to see something more novel /cutting-egde ; something more scientific than just an OpenCV put to good use.

Despite the questionable scientific "load", as a result of clear presentation of a good application, my recommendation is positive.

Author Response

(The authors gave the same response as above.)

Round 2

Reviewer 2 Report

The observations were adequately answered, therefore, this reviewer has no further comments in this regard.